# Using Augmented and Virtual Reality (AR/VR) to Support Safe Navigation on Inland and Coastal Water Zones

Tomasz Templin [1], Dariusz Popielarczyk [1,*] and Marcin Gryszko [2]

1   Department of Geodesy, Institute of Geodesy and Civil Engineering, Faculty of Geoengineering, University of Warmia and Mazury in Olsztyn, Oczapowskiego 2, 10-719 Olsztyn, Poland; tomasz.templin@uwm.edu.pl
2   INFEO Marcin Marek Gryszko, Królewny Śnieżki 18, 10-696 Olsztyn, Poland; marcin.gryszko@infeo.pl
*   Correspondence: dariusz.popielarczyk@uwm.edu.pl

**Abstract:** The aim of this research is to propose a new solution to assist sailors in safe navigation on inland shallow waters by using Augmented and Virtual Reality. Despite continuous progress in the methodology of displaying bathymetric data and 3D models of the bottoms, there is still a lack of solutions promoting these data and their widespread use. Most existing products present navigation content on 2D/3D maps onscreen. Augmented Reality (AR) technology revolutionises the way digital content is displayed. This paper presents the solution for the use of AR on inland and coastal waterways to increase the safety of sailing and other activities on the water (diving, fishing, etc.). The real-time capability of AR in the proposed mobile application also allows other users to be observed on the water in limited visibility and even at night. The architecture and the prototype Mobile Augmented Reality (MAR) applications are presented. The required AR, including the preparation methodology supported by the Virtual Reality Geographic Information System (VRGIS), is also shown. The prototype's performance has been validated in water navigation, specifically for exemplary lakes of Warmia and Mazury in Poland. The performed tests showed the great usefulness of AR in the field of content presentation during the navigation process.

**Keywords:** Digital Elevation Model (DEM); geo-information services; Mobile Augmented Reality (MAR); inland and coastal water zones; navigation; spatial databases and GIS; Virtual Reality Geographic Information System (VRGIS)

## 1. Introduction

The water environment is an unexplored world, which usually remains hidden. It forms part of the natural environment widely used for recreation, leisure, and tourism. Each year, countless people actively spend time on inland and coastal waterways using sailboats, motorboats, houseboats, or other transportation facilities.

Nautical tourism combines sailing and boating with vacation and holiday activities. It merges a variety of activities such as: travelling from port to port on a cruise ship, participating in sailing events such as regattas, chartering a boat and spending time on the water using the facilities (ports, marinas, restaurants, or entertainment venues). Inland water tourism supervisors do not require a license to operate a sailing yacht with a hull length of up to 7.5 m or a motor yacht with an engine power of up to 10kW for water tourism purposes.

People enjoying recreational water bodies support themselves with traditional paper maps or Personal Navigation Devices (PNDs) to navigate the water. Electronic devices with digital maps or dedicated nautical chart plotters are a barrier that makes navigating on shallow water reservoirs difficult for many users. Similar to most conventional navigation systems, they present navigation information in an abstract form, with arrows indicating the intended direction or as a "bird's eye view" map with the intended path [1]. However, navigation on the water is not as easy as car navigation and requires a good understanding

of the terrain. If this understanding is inadequate, the safety of sailors/motorboats is at risk and the vessel can be damaged, which may also have an adverse effect on the environment.

Augmented Reality (AR) technology has revolutionised the way digital content is displayed. By superimposing digital information directly on real objects, environments or maps, AR allows people to process digital and physical information simultaneously, improving their ability to absorb information, make decisions, and execute tasks quickly.

In recent years, AR technology has been popularised in various applications [2]. In geography, these include geovisualisation [3], spatial cognition [4], civil engineering [5], surveying and mapping engineering [6], environmental simulation [7], or education [8,9]. In tourism, AR is used to visualise places and activities before and during travel. It allows travellers to visit places of interest in a more meaningful way through digital overlays that contain interactive information about a place's culture or history [10–12]. In the world of cultural heritage, applications include virtual reconstructions of sites with the relocation of environment and objects [13,14].

Implementing AR for navigation is one of the most popular augmented reality applications in everyday life [15]. Using a mobile device camera combined with a Global Navigation Satellite System (GNSS), users can see their route against a real-world view. They can safely move from point A to point B based on GNSS position (autonomous or further improved by object recognition methods or Artificial Intelligence (AI) algorithms). In this process, the methodology of displaying navigation content to the user is becoming an important aspect. The use of AR has now become a subject of intensive research. In addition, this technology has had commercial implementations in the automotive sector. There is still work to be done to achieve an optimal interface and methodology that does not distract the user during navigation. As some authors have indicated [16], navigation in real environments with digital maps and augmented reality in mobile devices has caused some problems and is still challenging.

Safe navigation on the water requires access to up-to-date, reliable information on the shape of the bottom of the water body, location of dangerous places (shallows, stones, overhead power lines), navigable routes and information on the current situation on the water. They are obtained from various data sources and are usually stored in a digital database [17]. Geographic Information System (GIS) software is used to update and add new contents to the database, allowing for the analysis of the bottom shape, considering the current water level, identifying safe navigable routes, and marking dangerous places. Although the results are three-dimensional and digital, they are still visualised using traditional maps or bathymetric charts or displayed on two-dimensional (2D) screens. Although the 2D digital map can be very effective for navigational purposes, bathymetric presentations based on AR technology can be useful additionally for diving or fishing purpose.

Another element affecting the navigation process on shallow reservoirs is water traffic and the relationship between boaters. The real-time capability of AR can be exploited to present information about moving objects. In contrast to maritime applications, information identifying moving objects, the Automatic Identification System (AIS), is missing on inland reservoirs. Inshore, some systems monitor the situation on the water (for example—https://zegluj.mobi/mapa.html/, accessed on 10 March 2022) and keep their users' real-time position, which enables to use them for traffic management and socialising among the sailing community.

There are many applications available in the market for maritime and inland navigation. However, there is a lack of AR-based mobile applications dedicated to the general public. AR can add the advantage of true mobility and location awareness. It allows creating only those virtual objects necessary to supplement what a camera perceives (reality) [16]. This type of application is defined as Mobile Augmented Reality (MAR). It is an augmented reality-based technical solution dedicated to mobile devices (smartphone/tablet), Head-Up Displays (HUD), or optical Head-Mounted Display (HMD) [18]. It extends and enhances the experience of the mobile device user.

Only a small part of AR research focuses on water application. Bartolini et al. [19] describe the architecture of a novel infrastructure for coastal management with an innovative visualisation tool based on Augmented Virtuality. Mirauda et al. [20] presented a prototype Augmented Reality application specifically designed for monitoring water resources. The main goals were to provide an option to visualise 3D objects and real-time textual environmental data with an AR interface directly in the field.

Other authors concentrated on preparing the methodology of integrating and preparing products for VR and AR purposes using optical and acoustic data for underwater sites. An extensive review of the sensors and the methodologies used in archaeological underwater 3D recording and mapping is presented by Menna et al. [21].

Bruno et al. [22] present the methodology for preparing 3D semi-immersive or fully immersive underwater diving simulations. They use numerical products to prepare augmented diving services supporting divers in the navigation process inside an archaeological site. The tablet position and orientation are displayed like a 3D map representing the environment around the diver, information about underwater artefacts and structures.

There is a lack of research results in the literature on the feasibility of using AR to support users in inland waters. Most existing studies concentrate on coastal areas requiring advanced equipment and dedicated software.

Some research groups have started to employ augmented reality in water navigation on the sea. Many focus on implementing AR as innovative IT solutions to improve shipping safety, avoiding marine accidents caused by human error [23]. The primary goal of this research was to add new virtual contents to increase the number of observed vessels and add objects that are invisible but can affect the safety of navigation. For example, Bandara et al. [24] considered the feasibility of using AR to enhance visualisation in maritime operations to avoid collision in different environmental conditions. It showed that AR navigation lights, geo-locked to real physical hazards, can present navigational information in compromised visibility.

Hugues et al. [25] showed one of the first integrated solutions: a vision system and a thermal camera with augmentation. They conclude that the functionalities provided by AR must differ depending on people and weather conditions and require up-to-date contextual information. Oh et al. [23] proposed a navigation aid system based on AR technology that displays various overlaid navigation information on images from cameras to support swift and accurate decision making by officers. They verified their interface in the field and emphasised that further research on the AR interface to include a method for the efficient display and handling of information is needed.

The theoretical aspect of implementing AR for navigation was presented by Grabowski [26]. He presented several important research questions to introduce Wearable Immersive Augmented Reality (WIAR) systems in ship navigation. Procee et al. [27] focused on the user support aspect by conducting cognitive work analysis to derive a scientific base for a functional interface that best supports navigators in their work. To do this, they used Head Mounted Display (HMD).

Some authors analysed practical techniques used to visualise AR content in outdoor environments. For example, Hertel et al. [28] investigate the perceived egocentric distance of virtual objects in an open outdoor environment and how the visual attributes of coloration, shape, and relation to the floor influence the perceived depth.

The most compressive review of using AR for maritime navigation data visualisation was conducted by Laera et al. in two publications [29,30]. The first presents research and methodology for developing AR that assists sailing purposes. Their research concluded that current solutions for visualising sailing data have serious limitations and are technically outdated. In the second study, the authors investigate augmented reality technology in the field of maritime navigation. They assume that AR can be an evolutionary step to improve safety and reduce stress on board. However, the application of AR in the nautical field, specifically the maritime one, has not been sufficiently investigated. There is still much to be done regarding interface proposals.

Despite the rapid development of AR technology, the number of commercial implementations remains relatively small. Many Research and Development (R&D) projects show that due to the specificity of created products (handling only a limited number of tasks) and their various requirements (e.g., lighting conditions, techniques for superimposing objects), it is difficult to transfer the applied solutions to other applications directly. This is especially true for implementing AR technology outdoors in demanding, powerful, and hostile natural environment conditions.

The use of AR on water causes many technical difficulties. When navigating on water reservoirs, we usually encounter:

- Harsh environment in which the mobile application has to consider the water as an environment; weather conditions including extreme temperatures, intense sunlight, precipitation, and visibility obstructions; and strong movements due to waves (onboard ship) and in the case of motorboats' high speeds and accelerations.
- Obstructed conditions associated with the use of a positioning system. In many cases (especially near steep, wooded shore, narrow channels), some obstacles make it difficult to determine the GNSS position.
- The use of wireless networks for data transmission is often problematic due to existing bandwidth limitations or being completely blocked by a lack of signal coverage. It causes problems in real-time database access and affects the architecture of the mobile application.
- A ship on water is not static but moves, and its movements are heave, roll, and pitch (vessel motions). Therefore, navigation on water requires locating the mobile device, determining the ship's movement relative to the ground, and locating/moving the person (or device) to the ship.
- As in marine navigation, coastal navigation often lacks easy-to-interpret navigation marks or other visual reference points (problems for less experienced operators).
- A problem with handling electronic navigation devices is that they often have a complicated Graphical User Interface (GUI) that provides too much information. In addition, this information is not well organised, making users experience it as complicated rather than helpful.
- Optical viewpoint problem. For small water boats, the visibility of the water surface is relatively small because the optical viewpoint in the water body field is 1-2 m above the water level.

Uwe von Lukas et al. [31] additionally highlight issues arising from the use of AR in a maritime environment. They point out several factors contributing to the relatively weak development of technology in this area, highlighting relatively low R&D intensity, many small and medium-sized companies, and a conservative attitude of users.

However, the rapid advancement of technology results in today's mobile devices boasting fast, powerful Central Processing Units (CPUs) supported by Graphical Processing Units (GPUs). They have high-resolution screens, megapixel cameras, and various sensors such as powerful wireless data transmission methods, Inertial Measurement Units (IMUs), compass, and GNSS [32]. It enables them to provide a universal platform to prepare outdoor MAR apps dedicated to a wide range of users.

## 2. Materials and Methods

The survey of users' opinions (Section 2.1) shows that boaters and other water sports practitioners on inland and coastal waterways expect a simple, easy-to-use mobile app to improve navigation safety. It should offer a user-friendly interface that supports them without requiring knowledge of regulations, technology, and specialised equipment.

This requirement is satisfied by AR solutions. However, currently there is no representative standard of navigation information elements related to AR technology. This issue is the subject of many studies, and some research has investigated and tested end-users to define representative guidelines. The presented research is consistent with the above trend. It involved designing and testing an initial version of the visual AR interface for the

prototype MAR application. The following subsections demonstrate the results of a survey on user needs in this area, the concept proposal for an interface using AR on water, and its verification as the AR interface for a prototype MAR application.

### 2.1. The AR Interface User Needs Analysis

The authors aimed to determine the potential of AR technology for navigation purposes on the water by developing a navigational MAR application for inland and coastal water bodies and determine the usefulness of traditional navigation data sources such as bathymetric datasets, remote-sensing products, and 3D models as AR content. We also propose and implement a universal methodology for preparing AR content using its own VR solution.

The following assumptions were made as the starting point:

- The augmented contents will describe the part of the water reservoir where the user is positioned, help him reach the destination safely and more efficiently by augmenting the real world with navigation information, indicating the route, enabling efficient movement, and avoiding dangerous places.
- The app will offer technical opportunities to improve travel safety and help avoid dangerous navigation obstacles by alerting when approaching a dangerous area.
- GNSS will be the source of locations (track the user's location, geolocate other users or objects in the AR (e.g., obstacles a ship may encounter)).
- The application will present a navigation chart and user location/speed/direction, danger areas, other users, and visual appearance of 3D bottom models.

The MAR application should be widely available for mainstream phones and tablets. The MAR market has developed significantly in recent times. However, simple MAR apps available in application stores are still not widely adopted by consumers. Users consider these apps to be fun but not usable, and they quickly stop using them. That causes a need to prepare an easy-to-use, intuitive GUI based on UX design principles.

The authors conducted a survey addressing people active on the water to determine user awareness of AR in mobile navigational applications in 2021. The survey was implemented using an electoral form on 120 users of the Zegluj app. Answers are presented in Figures 1 and 2.

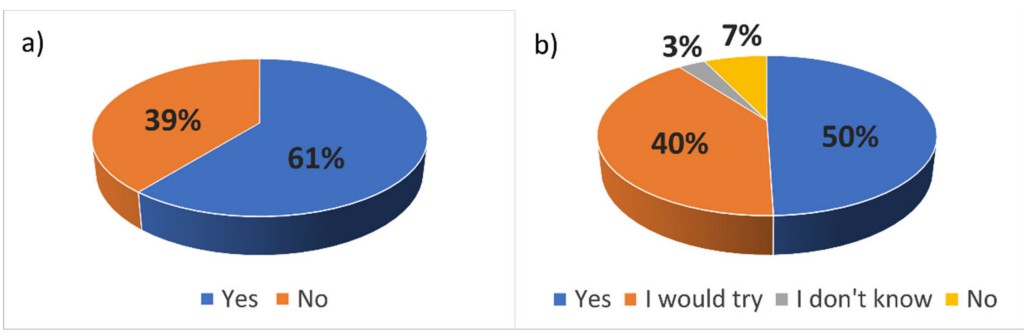

**Figure 1.** The evaluation of the need to implement AR functionality in a mobile navigation application: (**a**) Do you use an application supporting spending time by the water? (**b**) Would you use AR function in navigation application on the water?

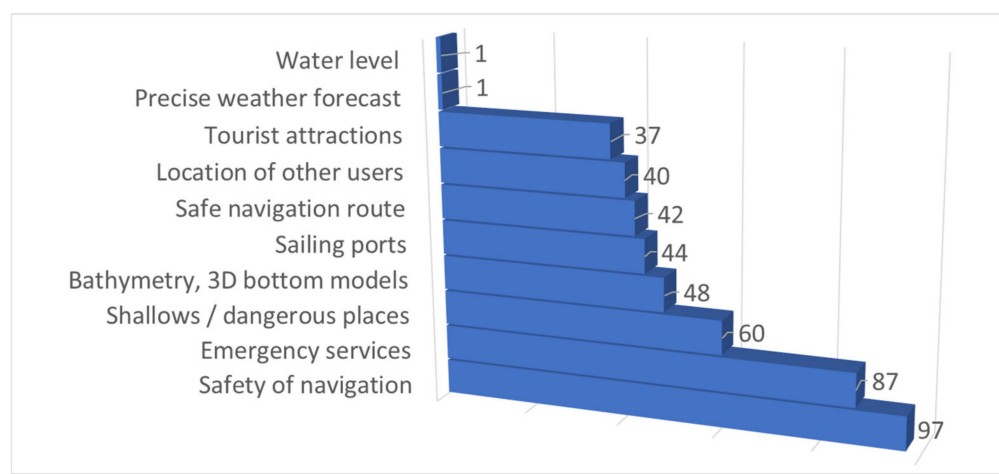

**Figure 2.** Identification of prioritised functionalities from the perspective of the mobile application user.

The analysis showed that people are aware of modern technology and use it to support their water activities (over 60%). Almost half of the respondents declared they will use the app with AR functionality, 40.2% would try it, and 3.1% had no opinion. Only 7.2% declared no need to use AR.

Safety of navigation is the most wanted feature where almost 97% of respondents consider information about shallow water and dangerous areas as important and 64% of them as very important. In addition, 87% recognise reporting a threat to emergency services as a needed function. Nevertheless, additional information such as bathymetry (48%), sailing ports (44%), other users (40%), and tourist attractions (37%) are also important.

In the following step, a list of potential elements to be displayed was analysed to determine the AR interface requirements of an on-water navigation application. Variables affecting the boat guidance process were defined. Of the information considered for display, the following were highlighted:

- Compass;
- Obstacles;
- Course;
- Boat route and speed;
- Position (longitude and latitude);
- Wind direction;
- Depth;
- Waypoints;
- Distance to waypoint;
- Estimated time of arrival (eta);
- Traffic information;
- Non-navigation zone;
- Course over ground.

This is the essential information for route calculation, monitoring, and primary checks of the boat status and navigation area. Due to the limitations of mobile device screens, the minimum set of parameters presented on the screen was selected from the complete list.

The preliminary selection of parameters intended to be displayed in the interface was made based on the authors' experience and a survey conducted among expert navigators and sailors. Three thematic categories relevant to amateur water sports practitioners were considered. This selection assigned an importance rate for every parameter displayed on the screen. The results are shown in Figure 3.

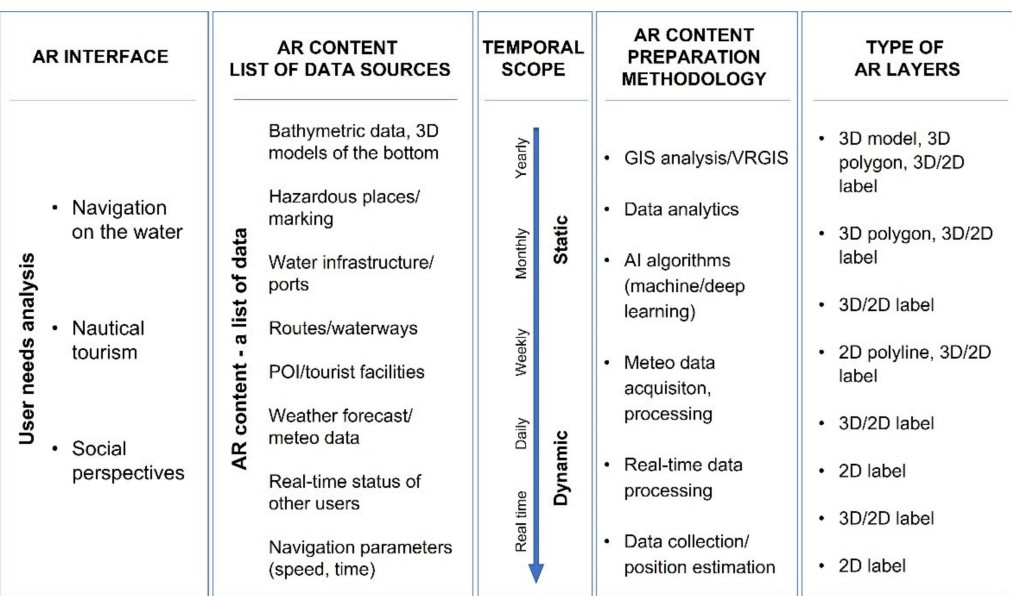

**Figure 3.** User needs analysis for visual AR interface.

Three application categories for AR interface were considered for ranking the required data: navigation on the water, nautical tourism, and social perspective. Potential data sources, temporal scope (needed frequency of update), geometric type of data, and data processing methodology for each category were also defined. The category of AR information is closely related to the type of activity on the water. In fact, these parameter sets may vary in importance depending on the specific task of the application.

The vast majority of the presented data relates to the navigation task on the water. It requires access to bathymetric data, dangerous places, markings of paths, and user travel routes. Nautical tourism involves accessing context-related information concerning the current position, including water infrastructure, tourist facilities, points of interest and weather data. The last category includes up-to-date information about boats to establish contacts, organise regattas, or use crowdsourcing data (e.g., monitor free moorings).

Analysing the scope of current research related to the use of AR on the water, it can be pointed out that the vast majority of it focuses on an interface designed for Head Mounted Devices (HMDs) used by professionals (trained individuals with experience in ship/boat driving). The authors' research concentrated on interfaces dedicated to individual consumers. They have little experience but are authorised to navigate through bodies of water on various vessels. The focus should be on identifying those elements that are important to them. The four types of AR functionality proposed to be implemented in the application are shown in Figure 4.

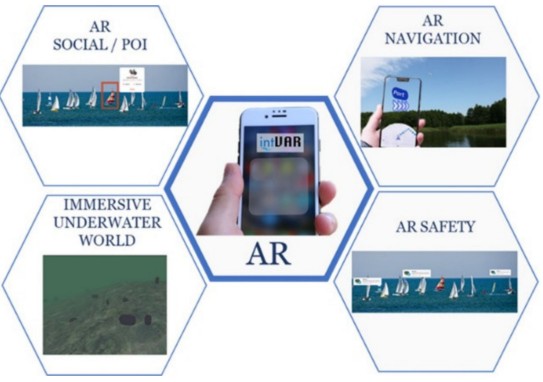

**Figure 4.** Types of AR functionality supported by the proposed architecture.

## 2.2. Architecture of the System

Traditionally, mobile geo-application refers to popular applications and frontiers, including Location-Based Services (LBS), social media, and Augmented Reality (Fu 2015). LBS refers to information services that integrate the location of mobile devices to provide added value to the user. The social media features offer solutions that can help users communicate easily. AR combines data from different sources with data from the human senses.

LBS is an up-and-coming technology to deliver valuable services. It is the foundation for developing outdoor MAR applications. An LBS requires five essential components: the service provider's software application, a mobile network to transmit data and requests for service, a content provider to supply the end-user with geo-specific information, a positioning component, and the end-user's mobile device. The most crucial element is the positioning module that provides information about the current location. Nowadays, GPS, Galileo, GLONASS, BeiDou satellite positioning systems, and EGNOS are used. They provide a submeter accuracy of position determination [33,34].

The integration of GNSS and AR technologies offers new technical opportunities to improve travel safety and to help avoid dangerous navigation obstacles. In order to display AR layers, an appropriate system architecture is needed to deliver the content to MAR app users.

The proposed architecture is based on several interconnected building blocks. Each of them is responsible for different tasks. The typical architecture of a location-based mobile augmented reality system is used to acquire data about the observed world, acquire AR content, and overlay it on the viewed scene. The principle of operation and overall architecture of the proposed AR inland and coastal water application highlighting the major modules and their relationships are shown in Figure 5. The diagram consists of the following modules:

- Tracking module—module responsible for obtaining geographical position and orientation of the mobile device based on the sensors (GNSS receiver, accelerometer, gyroscope).
- Interaction module—the subsystem that allows users to interact with electronic devices (AR interface).
- Presentation module—the subsystem responsible for displaying content using AR layers (3D objects, polylines, 2D/3D labels, pictures).
- A world model—AR content, a database containing information describing the real world, including data about presented objects and methods.
- Communication module—communication channel between a client and server, the component responsible for sending/receiving data into a database.
- Filter module (data filtering)—a module responsible for selecting AR content based on preferences set by the user.
- Search engine—the module that selects content based on a query generated by the client.

In the proposed solution, the architecture is based on a 2-tier client–server architecture. Client–server architecture is a computing model in which the server hosts, delivers and manages most of the resources and services to be consumed by the client. This type of architecture has one or more client computers connected to a server over a network or internet connection.

In this model, tiers realised the following functions:

1. Client-side—responsible for gathering information from the user, sending the user information to the business/data services for processing, receiving the server processing results, and presenting those results to the user.
2. Server-side—responsible for receiving input from the client. It interacts with the data server to perform operations that the application was designed to automate (AR

content, weather forecast, DEM's of the bottom). It sends the processed results to the client side.

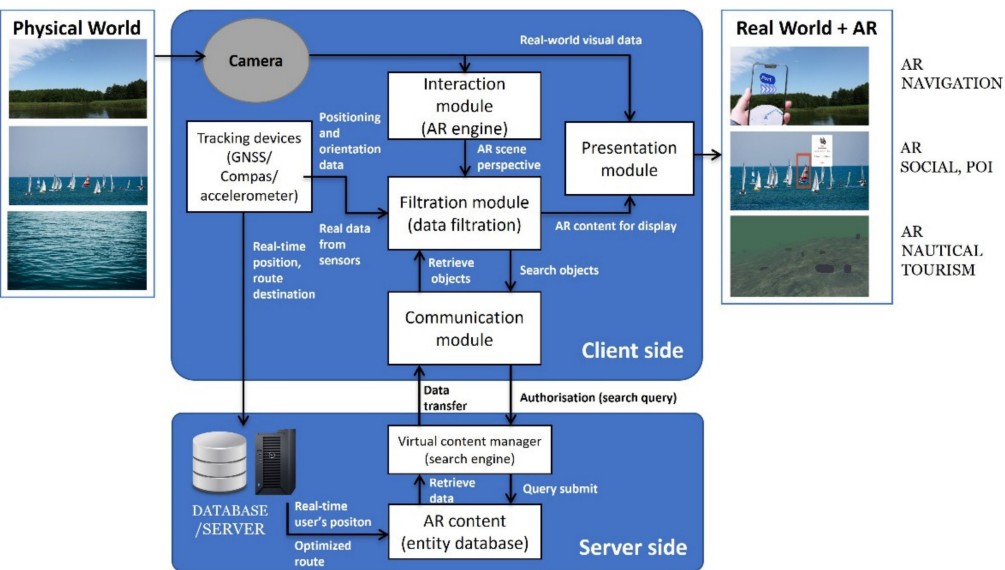

**Figure 5.** The overall architecture of the proposed AR shallow water application highlighting the major modules and their relationships.

In a typical situation, the communication is initiated by the client. The client obtains the position and orientation of a mobile device based on the tracking device module. Based on the parameters set by the user interface (interaction module) and calculated relative pose, the database query is generated, and objects captured by the mobile device are identified. After recognising objects, the subsystem returns object IDs, camera position, and orientation for each object. The client downloads this data into the presentation module and displays it.

*2.3. AR Content*

Augmented reality content is computer-generated input used to enhance parts of a user's physical world via smartphones, tablets, or smart glasses. Users can generate this content directly or use previously prepared materials. AR content is sometimes delivered as 3D models or visual, video, or audio content.

Section 2.1 identifies the sets that constitute the visual elements of the AR interface (AR layers). A potential data source and a methodology for processing it to deliver the required AR content has been identified for each of them. A detailed list of AR layers is provided in Figure 6.

A critical source of data for users on the water is bathymetry. In reality, the helmsman cannot see the shape of the bottom. He can only see the water's surface. Bathymetry information is not just a curiosity, but a critical element affecting the safety of navigation. Based on the shape and morphometry of the bottom, it is possible to determine areas that are dangerous to navigate (shallows), rocky reefs, single boulders that can damage yachts, motorboats, and houseboats, as well as sudden dips and shallows, which are dangerous for swimming, especially for children and teenagers, but attractive for anglers and divers.

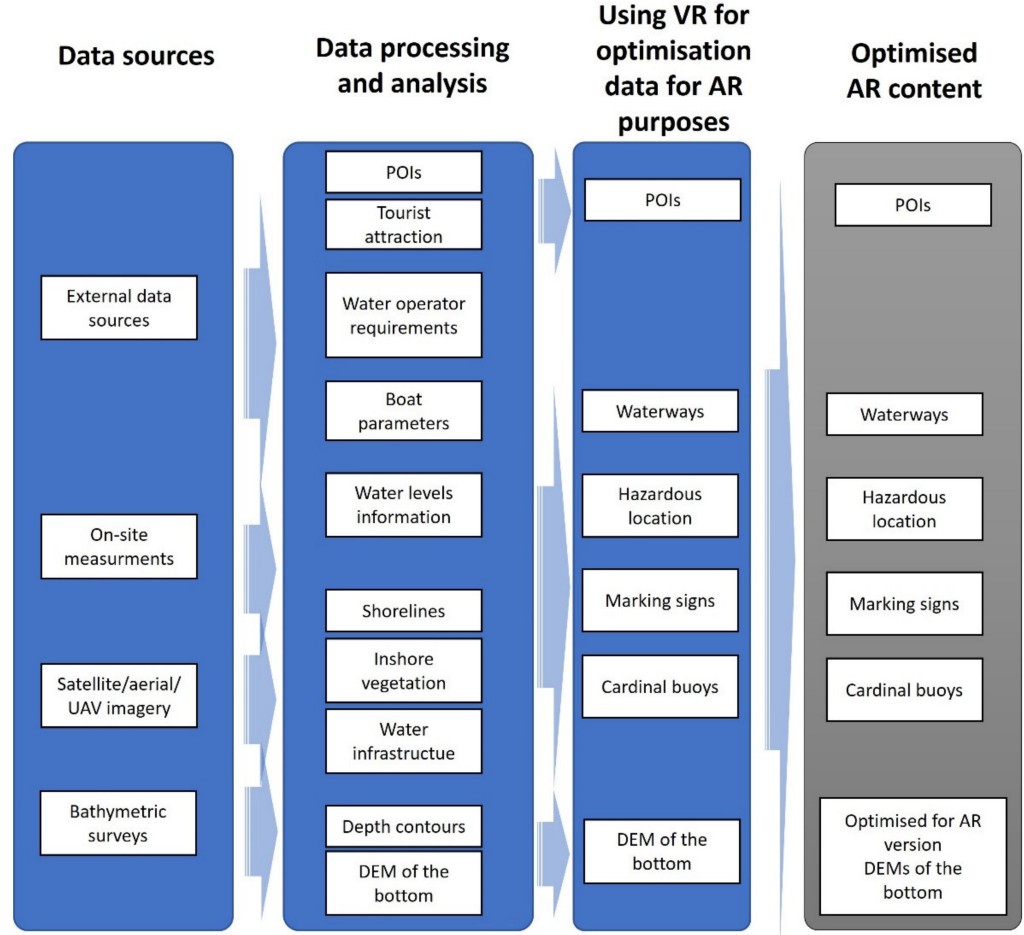

**Figure 6.** Data sources and processing methodology for preparing AR-optimised layers.

## 3. Measurements and Validation

### 3.1. Study Area

Our chosen test area, the Great Masurian Lakes, is a beautiful region often called the Land of Thousand Lakes. The selected area is located in the northeastern region of Poland, known as the Masurian Lake District (Masurian Lakeland). It is the most popular region among sailors. This fashionable region is visited by about 1 million Poles and 200 thousand foreigners every year. In the summer, only in Masuria, about 50 thousand sailors and motorboaters (about 10 thousand yachts) spend their free time actively every day.

Generally, water covers more than 70 percent of the earth, but more than 80 percent of its surface remains unexplored. Unfortunately, many reservoirs in Poland have shallows with rocks, reefs, and fallen trees, dangerous for sailors. Dangerous places make marine navigation very difficult. Therefore, all shallow water areas should have up-to-date bathymetric charts to ensure the safety of shipping lanes [17,35–37]. Nowadays, high-resolution multibeam echosounders are used more commonly. They allow not only the obtaining of a very accurate bathymetry of the shape of the tank bottom but also of its characteristics and structure. Multibeam systems also identify underwater objects and obstacles (rocks, wrecks, sunken trees).

For the feasibility tests of the prototype version of the MAR app on water reservoirs, two lakes have been chosen: the biggest lake in Poland, Śniardwy, and one of the longest, Bełdany. The first step to building the database for the MAR app was to perform bathymetric measurements.

### 3.2. Bathymetric Surveys/Data Analysis

Experiments were realised on two sites (Figure 7) with different morphology: the largest Polish lake, Śniardwy (July 2019), and post-glacial lake, Bełdany, one of the most popular among sailors (November/December 2019). These lakes were measured using various hydroacoustic technologies. Lake Śniardwy was measured with a single-beam echosounder (SBES) based on profiles spaced 10–50 m apart. Lake Bełdany was measured with a high-resolution multibeam echosounder (MBES), giving several points per square meter.

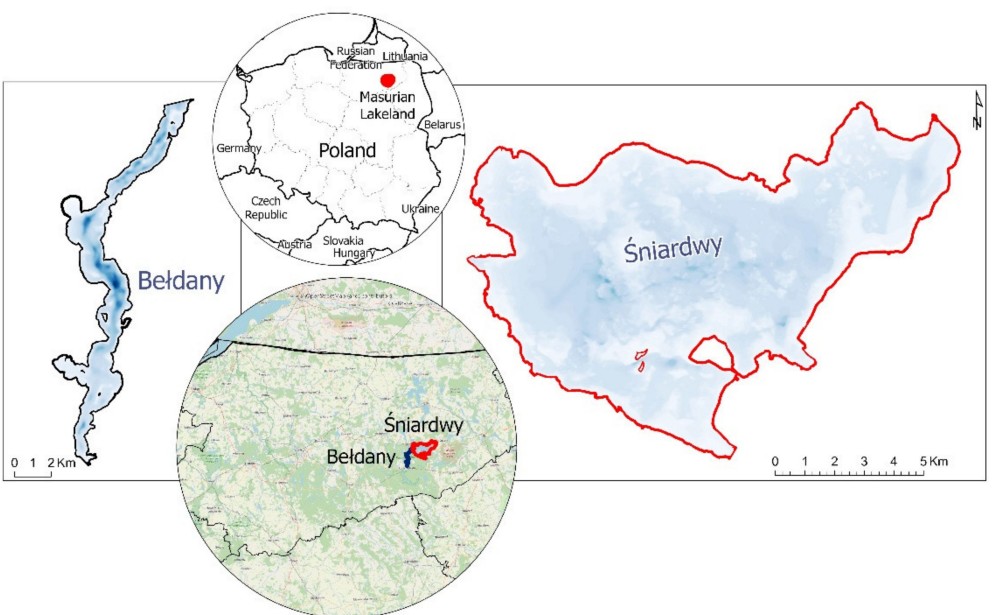

**Figure 7.** Study area.

AR content for the MAR app was prepared by carrying out the following task. First, an on-site survey of hazardous areas was made for this area, their horizontal location was identified, and bathymetry was analysed. As a part of the survey work, GNSS and hydroacoustic techniques were used to verify the shape of the lake bottom and identify shallow, hazardous areas. The work was then continued using GIS software (ESRI ArcGIS Pro) and self-made VRGIS software for bathymetric data analysis.

Professional hydrographic surveys of selected parts of both lakes were carried out to identify dangerous places (shallows with rocks). Lake Śniardwy was measured with an SBES Simrad EA501p single-beam echo sounder, while Lake Bełdany was measured with a Reson T50P multibeam system. In the next step, based on a numerical terrain model of the bottom, shallow places dangerous for navigation were determined. Computer simulation of water level lowering was used, taking into account bathymetry and long-term average water levels in the lake (Figure 8).

In the next step, we elaborated a design to place cardinal buoys (IALA system: International Association of Marine Aids to Navigation and Lighthouse Authorities) to assist sailors in avoiding obstacles. The marking was implemented after direct verification of correctness and water rescue services (Figure 9). Finally, a navigation lanes layer was developed using the developed bottom models and identified hazardous areas (safe and reliable). These routes indicate safe waterways using side marker buoys.

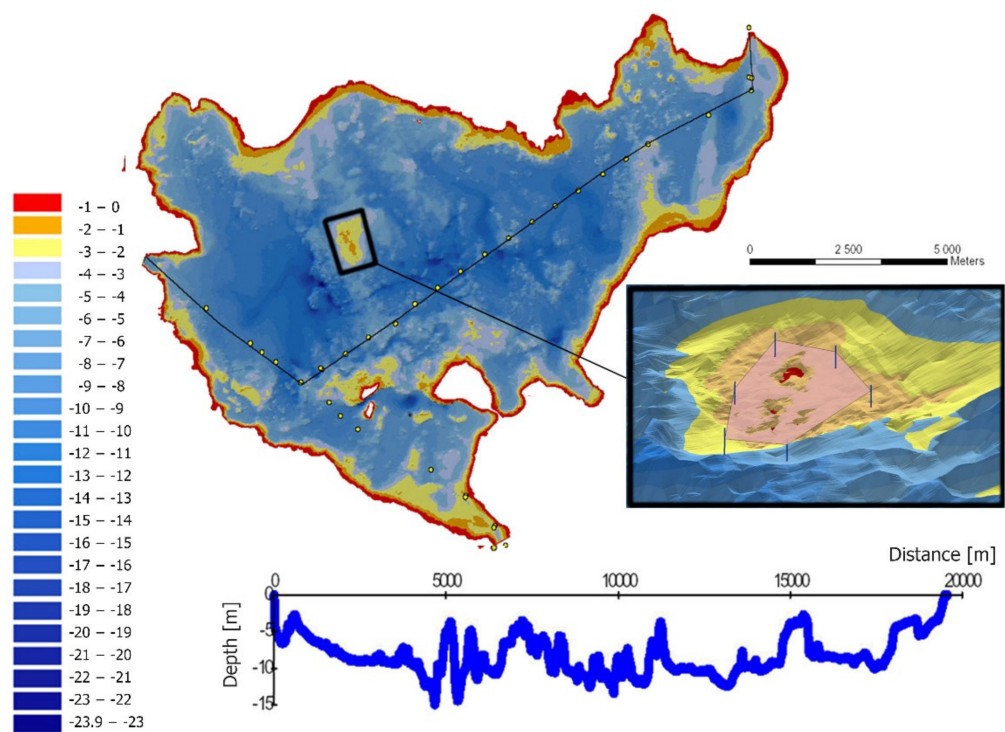

**Figure 8.** Marking the navigable route on the largest lake in Poland, Śniardwy Lake, with the depth profile along the route. Inset map presents a sample of cardinal buoy location (IALA system) used to mark hazardous locations obtained from GIS analyses.

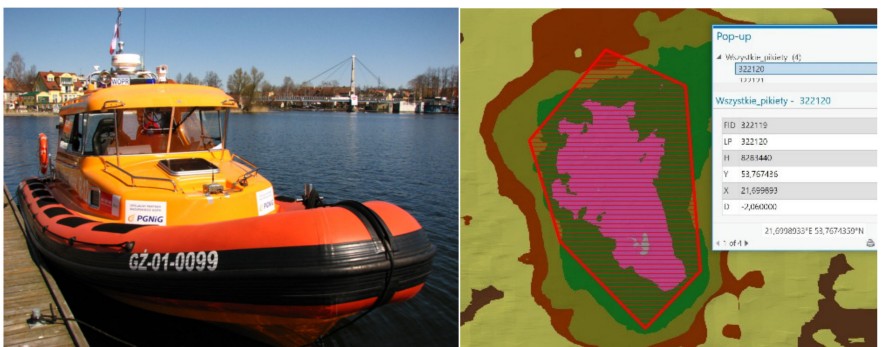

**Figure 9.** On-site verification of cardinal buoys (measuring platform—Water Rescue Service boat (**left**), the model with depth readings (**right**)).

GIS analyses applied in the data processing allowed identifying dangerous places, cardinal signs, and buoys for their marking. GIS software allows for 3D visualisation of the obtained results on the monitor screen. This visualisation method offers the lowest immersion level, resulting in limitations to the visual assessment of the validity of the obtained findings. It also does not control the optimisation procedure for further use of prepared content in the MAR application.

In order to optimise the procedure of data adaptation to the needs of the prepared mobile application, additional software was used to increase the level of immersion through the use of VR. The dedicated, self-made VRGIS software was used. It is based on the Unity game engine (https://unity.com/, accessed on 10 March 2022) and allows importing GIS data, visualising 3D models (TIN, GRID, textures), performing analysis, and adding and verifying hazardous areas. In order to verify the already-prepared data, it was further analysed using VR Oculus Rift glasses with motion controllers and gesture manoeuvres. The

methodology of verifying hazardous area signs based on DEM of the bottom is presented in Figure 10.

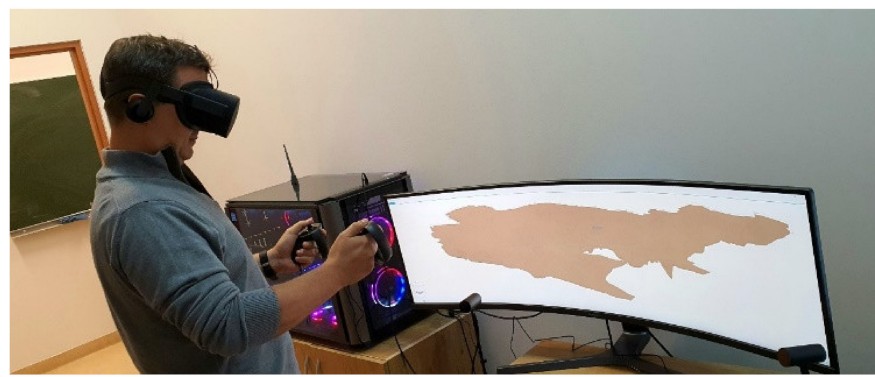

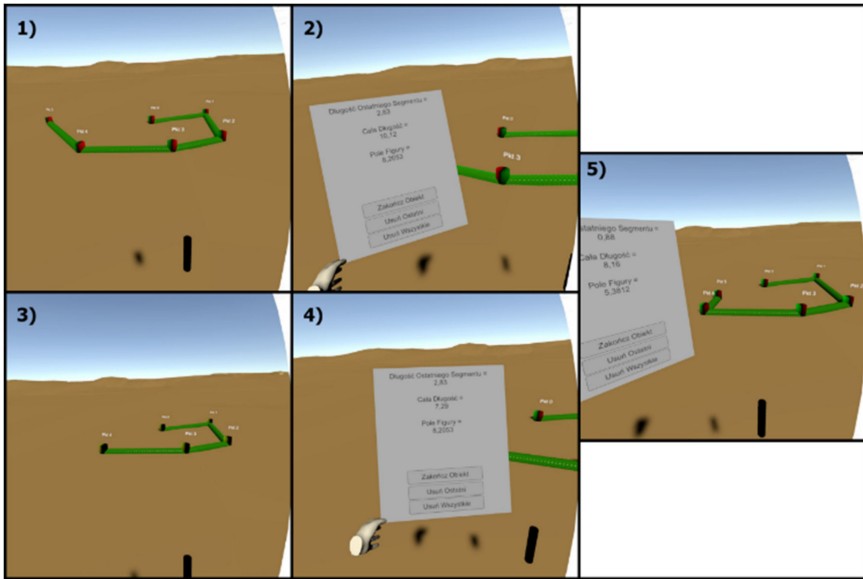

**Figure 10.** Optimisation of marking location and verification of hazardous area signs based on DEM of the bottom using self-developed VRGIS app. Screens from Oculus Rift with motion controllers: (**1**) The new point creation. (**2**) Settings new attributes for created point. (**3**) Removing existing points. (**4**) Display all values after deleting one point. (**5**) Display all values after adding one point and refresh the display.

The proposed methodology optimised AR and prepared the Śniardwy and Bełdany Lake DEM bottom model by adjusting the number of vertices, spatial distribution, and texture overlay verification. In addition, the marking of waterways and dangerous areas layer was also loaded as a text file and then it verified the spatial distribution of the markings and the correctness of the mapped waterways.

### 3.3. Prototype of MAR App

3.3.1. MAR App Functionality

The main element of the MAR mobile application is a unique functionality using AR technology. The key functionality of MAR is a visualisation of a route on the water, the identification of obstacles which are difficult to perceive, or the display of information on the water reservoir directly where the phenomenon is taking place.

The application will add computer-generated content to the observed (through a camera built into the device) real world. The augmented contents will describe the part of the water reservoir showing the route, enabling efficient navigation, and avoiding dangerous areas. Augmented reality in water navigation will support safe travel on the

water and guide the user to the specified destination. The integration of GNSS and AR technologies offers new technical opportunities to improve travel safety and to help avoid dangerous navigation obstacles.

In order to guide the skipper along a specific path, it was decided to use an interface similar to AR car navigation. The research tested different abstract shapes such as lines and planes and different 3D objects such as arrows and 3D object models.

### 3.3.2. MAR App Implementation

To realise the assumptions made and test the effectiveness of displaying objects on water, a prototype MAR app for iOS was implemented and performed all the necessary tests. Finally, for the app dedicated to navigating in a water environment, we decided to use:

- 2D labels (descriptions);
- 3D labels;
- Abstract shapes (boats moving on water);
- Watermarking symbols;
- Realistic 3D objects (such as virtual buoys rendered on the water surface);
- DEMs of the bottom.

Examples of implementation of the algorithm for superimposing selected objects (2D labels, 3D objects) based on location and data defining the orientation of the smartphone are shown in Figure 11.

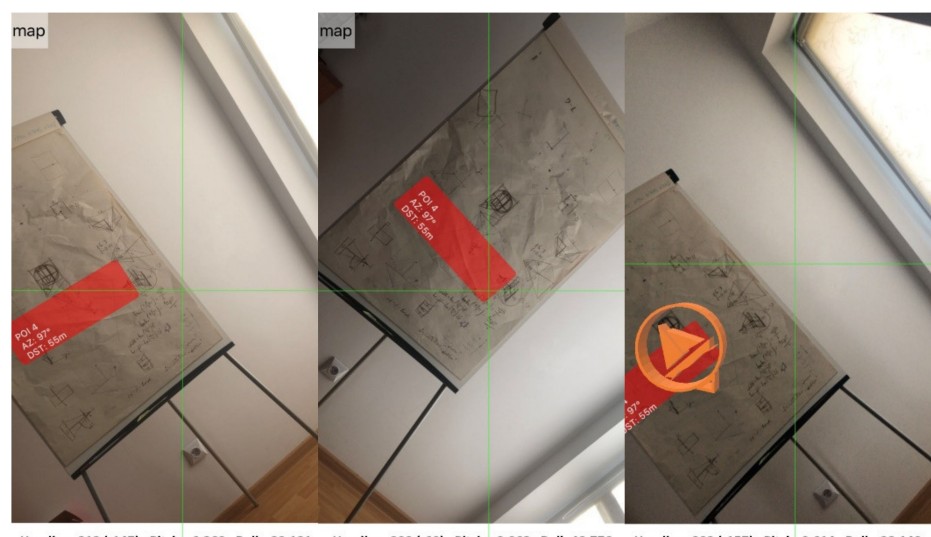

**Figure 11.** Test application presenting labels and 3D models of spatial objects based on sensor readings.

The proposed solution is based on a location-based approach to AR. In this solution, it is essential to define rules for displaying objects' visibility priorities in the recorded real-world view. Using latitude, longitude, and altitude data, the app computes the relative location of target Points of Interest (POIs) and displays them correctly on screen. After determining the location of the object, graphical symbols with interactive annotations are overlaid on top of the captured image.

Due to the potentially increasing number of objects to display, a unique mechanism is introduced to consider the spatial relationships. In the prototype of the MAR app, an algorithm was proposed to optimise the displayed content. It utilises distance threshold and direction from the camera sensor to the objects to determine how many will be overlaid on top of the captured image. It allows the filtering of objects and visualising of only part of them, not too far from the device and inside the active field of view.

The implementation requires combining mobile device position and orientation (determined with GNSS receiver, IMU, and magnetometer) with geographical objects using spatial relationships. The approach considers three aspects of the problem: distance, angle, and the assigned priority determined by these parameters (order) (Figure 12).

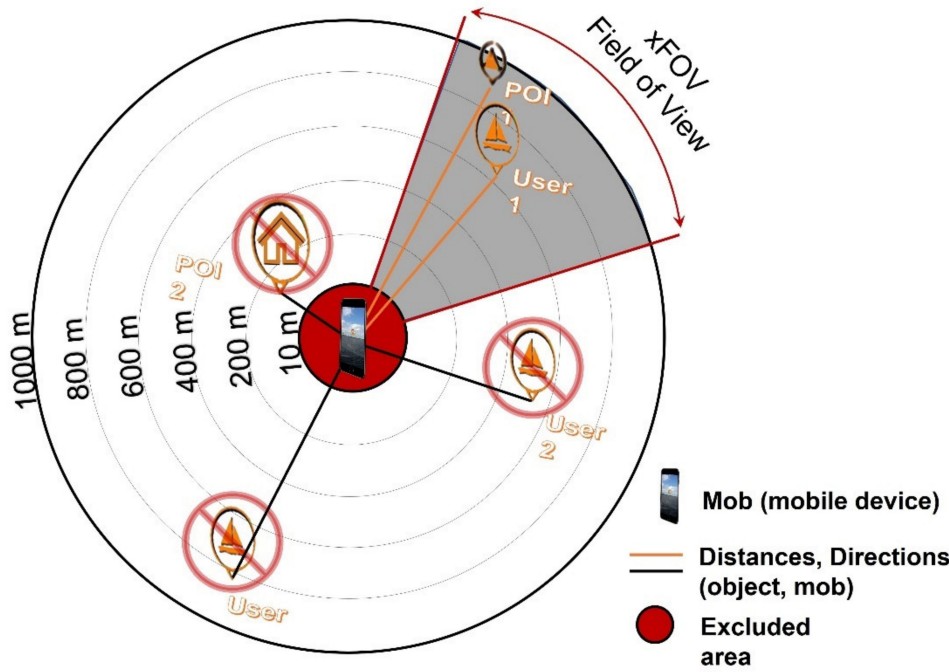

**Figure 12.** Concept of AR content matching in dynamic water conditions. Geographic objects are defined by distance criterion (5 distance thresholds), horizontal directions, and actual horizontal field of view.

When the mobile device sensors determine the position and orientation, a database query is generated to check the objects of interest and retrieve the GNSS coordinates of features meeting the distance criterion (up to 1000 m). In the next step, the list is further filtered based on the distance assumptions calculated between the visual sensor and the geographical objects and the orientation of the mobile device. The methodology is shown in Figure 13.

The acceptable distance criterion represents the value of the distance between the visual sensor and the geographical object (minimum and maximum). The spherical distance is calculated using the haversine formula Equation (1).

$$distance_{mob\_user1} = 2R arcsin \sqrt{\left(sin\frac{Lat_{mob}-Lat_{user1}}{2}\right)^2 + cos(Lat_{mob}) \times cos(Lat_{user1}) \times \left(sin\frac{Lon_{mob}-Lon_{user1}}{2}\right)^2} \qquad (1)$$

where:

- R is the radius of the Earth (km);
- $Lat_{mob}$, $Lat_{user1}$ are the latitude of mobile device and latitude of point1 (user position);
- $Lon_{mob}$, $Lon_{user1}$ are the longitude of mobile device and latitude of point1 (user position).

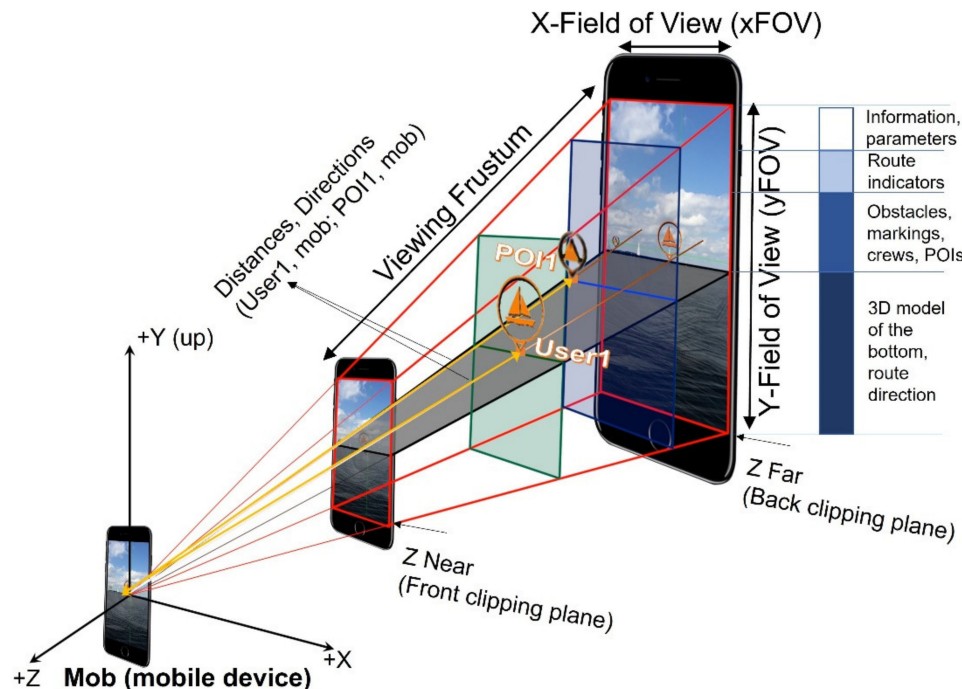

**Figure 13.** A visualisation of a real-world 3D coordinate system, determined by the viewing frustum created by the visual sensor, with the origin at the centre of the visual sensor. The X and Y axes are parallel to the screen. The Z-axis, which corresponds to the negative orientation direction of the visual sensor, is perpendicular to the screen. The AR layer layout concept for the MAR app prototype is on the right.

In the next step, the aspect of orientation is analysed. A direction is calculated to determine whether the object is within the actual horizontal field of view. It determines whether the object is inside the field of view and should be displayed as AR content. The direction value is calculated according to Equation (2).

$$direction_{mob\_user1} = arctan\left(\frac{\begin{array}{c}cos(Lat_{mob}) * sin(Lat_{user1}) - sin(Lat_{mob})\\ *cos(Lat_{user1}) * cos(Lon_{user1} - Lon_{mob})\end{array}}{sin(Lon_{user1} - Lon_{mob}) * cos(Lat_{user1})}\right) \times \frac{180}{\pi} \quad (2)$$

where:

- $Lat_{mob}$, $Lat_{user1}$ are the latitude of mobile device and latitude of point1 (user position);
- $Lon_{mob}$, $Lon_{user1}$ are the longitude of mobile device and latitude of point1 (user position).

The calculated distances and directions are compared with the corresponding thresholds and any geographic object that does not meet the requirements is not displayed. Additionally, each object has been assigned a priority (the category of the object). They define the label/3D model's size and colour of the object depending on its type. In the prototype version of the application, five categories based on distance (200 m, 400 m, 600 m, 800 m, 1000 m) have been proposed.

## 4. Tests and Results

An AR inland and coastal water navigation system prototype was developed based on the architecture described in Section 2.2. Our solution is intended for the entire community of shallow-water navigators, mainly for the inexperienced people whose ability to note

relevant information on the water called 'seaman's eye' is insufficient [38]. A simple smartphone app could simply serve in providing added certainty for novice sailors.

Using the prototype MAR app, simulated and real application experiments were conducted. In the experiment, each AR functionality of the prototype was tested (on the water), the applicability in varying locations on the boat was evaluated, and the suitability as a navigation device was assessed. The ability to display selected AR layers and their usefulness in navigation was tested.

There are several categories of augmented reality technology, each with different objectives and applicational use cases. For the prototype of the MAR app, the most popular types: marker-based AR, markerless AR (sometimes called location-based or sensor-based AR), superimposition-based AR, and projection-based AR were explored.

Marker-based augmented reality (also called Image Recognition) uses a camera and a visual marker, such as a QR/2D code, to produce a result only when a reader senses the marker. The markerless AR uses a GNSS, digital compass, gyroscope, or accelerometer embedded in the device to provide data based on position.

Projection-based augmented reality works by projecting artificial light onto real-world surfaces. Superimposition-based augmented reality either partially or fully replaces the original view of an object with a newly augmented view of that same object.

The first tests were performed to explore the suitability of different augmented reality approaches in the water environment. The recent research shows significant problems with using native AR Software Development Kits (SDKs) for water surface detecting. For example, Nowacki et al. [39] tested the potential of the ARCore and ARKit platforms on different surfaces and under different conditions. They concluded that the water mapping sheet was disappointing in both cases (ARCore, ARKit), but ARCore detected many more characteristic points.

The first test investigated the feasibility of using water surface recognition with the popular ARKit and ARCore solutions. The tests were conducted during experiments on two Great Masurian Lakes—Śniardwy and Bełdany. Unfortunately, the mobile device could not calculate the world origin (centre and orientation of the reference system where other objects are located) or find any tracking surface. Another test on Lake Bełdany under different weather conditions was conducted. The lake was calm, and the water surface was flat that day. The device calculates the world origin and tracks the surface only when the shore is partially in the camera's field of view. When the device's camera moved so that only water was in the field of view, the origin and all 3D objects placed in the scene began to float due to the lack of tracking points.

The results showed that a different approach should be taken, omitting the tracking of the world and focusing only on the device's sensors. With more accurate sensors integrated into new smartphones, this approach (location-based) has become increasingly important and gained popularity, especially in the application area of personal navigation; storytelling; Architecture, Engineering and Construction (AEC); cultural heritage; and tourism.

Further work assumed that AR objects were at the same height as the device on the water. This assumption eliminated the problem of low accuracy in determining altitude using GNSS methods. A test of application that presented labels and 3D objects in the device view based on a horizontal position, compass, and accelerometer was developed.

Due to technical capabilities, it was developed on the iOS platform using the native ARKit platform. The AR content is superimposed as a 3D object, 2D/3D labels, polylines, or polygons on the real world obtained from the smartphone camera. All AR objects are displayed on a smartphone screen. There are three information elements represented simultaneously in this solution: DEM of the bottom, markings, and traffic information.

The solution presents three groups of AR information elements. The route is indicated by a line between buoys/cardinal marks to provide the user with the necessary information about navigable routes and dangerous places. The DEM of the bottom shows depth and shallowing information as a 3D model. 3D objects show other users' locations with information about them (safety and social aspect) and selected POIs.

The information elements change size and colour depending on the distance from the boat to improve distance perception. Some of the described AR objects are shown in Figure 14.

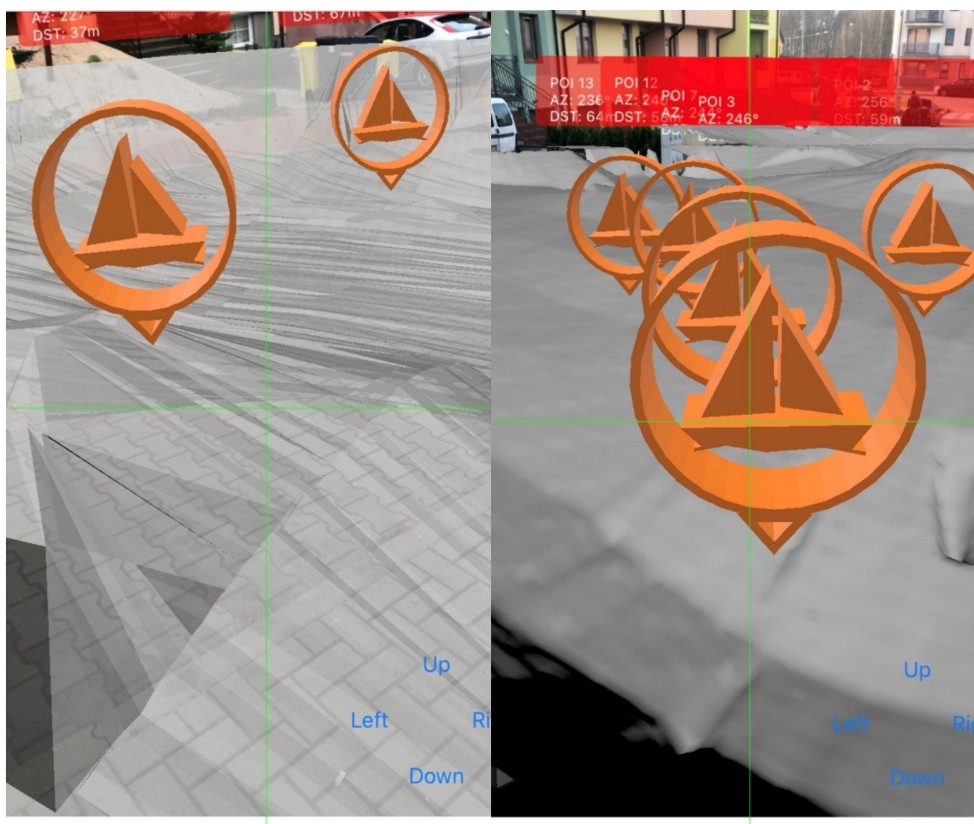

Heading: 121 (121)  Pitch: -28.617  Roll: -4.232    Heading: 116 (116)  Pitch: -24.100  Roll: 0.452

**Figure 14.** Examples of Augmented Reality applied to the navigation process—data fusion of the virtual location of sailors and Digital Elevation Model of the bottom of the water reservoirs.

In the following step, tests were conducted using the prototype MAR application on real data. The results are shown in Figure 15, which shows other users' locations whose position comes from the Zegluj navigation system (http://zegluj.mobi, accessed on 10 March 2022) and selected POIs. The tests showed that location accuracy did not matter when presenting distant objects. Some problems occurred with objects located at distances lower than 200 m. AR 3D objects "shifted" into the inaccurate position.

Further tests were conducted to reduce the impact of the inaccurate position on AR display. Changes were made to the code to optimise the frequency of refreshing the position depending on the travelled distance. In addition, changes have been made to improve the accuracy of the user device's location, and new filters have been introduced to optimise the determined position.

Tests were conducted on an iPhone 8. Approximately 50 points with 3D icons and the DTM model of Lake Śniardwy were rendered. The results were auspicious, and there were no rendering problems. The application worked very smoothly.

The movement through the digital terrain model while driving was also tested. When the device was in motion, location inaccuracies did not affect the presentation or immersion in using the application. The impact of inaccuracies in location determination was noticeable only if the device did not change its location at all. However, the frequency of position and orientation refreshing should depend on the distance, eliminating this problem.

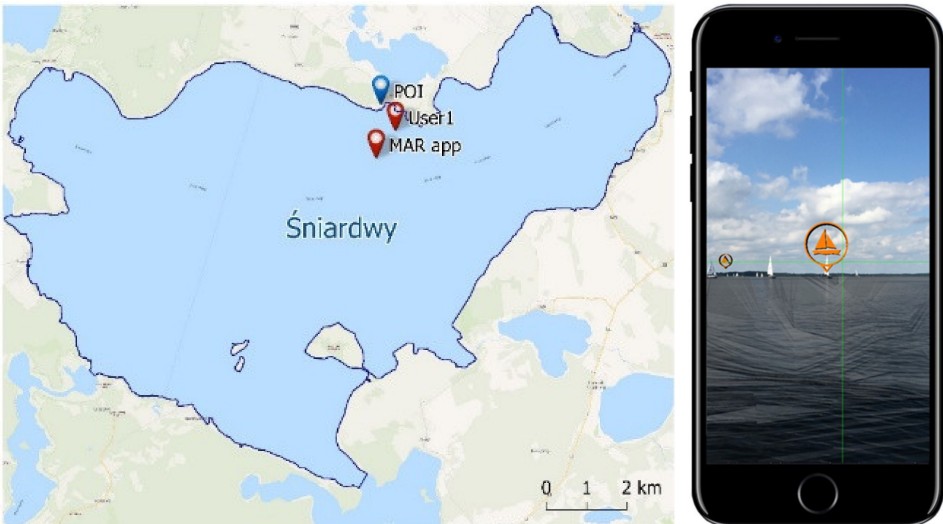

**Figure 15.** The screenshot of the prototype of the MAR application (**right**). The AR visual interface presents real-time AR content about other boats and POIs (**left**).

## 5. Discussion

One of the main goals of this work was to verify the usefulness of AR technology for navigation in inland and coastal waters. In this research, the authors explored the feasibility of mobile devices as a hardware platform for building an AR-based mobile application. A visual AR interface was proposed for a general user consistent with the concept of nautical tourism, with thematic categories defining the range of content presented on the application screen. Due to the issue's complexity, it is rather a voice in the discussion, showing a methodology in preparing data for an AR application, a potential system architecture supporting a mobile AR application, and the proposition of an AR interface dedicated to this issue.

Existing solutions focus on implementing AR within professional, costly, and dedicated marine vessel solutions. At the same time there is a large group of sailors with less experience navigating on challenging sea and shallow areas. There are also professionals who sail recreationally in shallow waters. Despite much research, there is a lack of unification within a universal approach. However, the authors agree that the critical element is to work on an optimal AR interface and to produce AR context-related data dedicated for onscreen display. This paper examines these issues from the perspective of the user, who expects to be supported in the process of safe navigation on the water as well as contents that facilitates activities on the water. The importance of the presented results using AR for inland and shallow zones navigation thus lies both in their generality and relative ease of application to new areas.

The presented prototype AR application is the first approach that will be further developed. The work identified several issues and difficulties in implementing AR on mobile devices. However, the obtained results correspond with those by authors of other studies, who indicate that AR can be a key technology supporting water users. The experiments show that this technology has the potential to show a model of a body of water, to help efficient navigation on the water by augmenting the real world with navigational information, indicating the route, and avoiding dangerous places. Additionally, it can provide social information about other users, places worth visiting, and tourist infrastructure. Therefore, it can significantly impact the development of nautical tourism, becoming a source of new innovative tourism services.

One of the main limitations of the proposed solution is the hardware itself. For a mobile device to work efficiently, it must be placed appropriately on the boat, providing a camera view in front of the subject and a clear view of the screen by the captain. To simplify the research, the authors did not address the topic related to the safety of using

this solution, the boat user's control, and the simultaneous observation of the screen of the mobile device. These topics are the subject of many studies. Several authors point to the need for HMDs in this area. It seems that a big opportunity is the introduction of HUD technology, which is especially evident in the automotive industry. There, the display of AR content on the windshield (HUD) is becoming more common in navigation systems in cars to minimise visual interference for the driver while providing relevant information while driving. Work is also currently underway to apply similar solutions to water navigation. They aim to display information for the skipper, steering the motorboat efficiently.

It is challenging to implement augmented reality on water. For most algorithms, it is difficult to achieve both good accuracy and high efficiency. Sensor-based methods can achieve reasonable efficiency, but their performance is often limited by the low precision of sensors used in mobile devices. Vision methods typically require significant computational and memory space to process an image consisting of many pixels. They require matching the image to a large database and estimating the geometric transformation between the captured image and the recognised object from the database.

A hybrid approach that integrates vision-based and sensor-based methods can potentially combine their complementary advantages, but its implementation is still a complex task and requires great knowledge. The authors have already conducted several experiments in optimising position and orientation estimation, position filtering, and optimising its display.

The performance of mobile devices is one of the most critical factors in implementing AR technology. Other important factors are low power consumption and memory management. Although the performance of mobile Central Processing Units (CPUs) has increased significantly and Graphics Processing Units (GPUs) can speed up calculations, the memory of today's smartphones is still not enough for performing some advanced operations (such as object recognising). Implementing the client–server model is a possible solution. It allows realising some operation on the server-side or cloud via the Internet. However, the network connection is not accessible anywhere, especially in rural areas with low population density. Due to these reasons, it is necessary to consider the functional requirement for the system.

A system architecture is a conceptual model that defines the structural behaviour of a system. It could be described as a set of significant decisions about the organisation and components of a system solution.

The proposed system architecture was defined based on the recent methodologies. Nevertheless, the complexity of the proposed architecture prevents the proposed solution from fully meeting all of the requirements.

The most popular mobile systems are installed on various hardware platforms provided by different vendors. The MAR app should work on devices with different screen sizes and resolutions, which brings on technical risks for the app. It requires determining different testing strategies and analysing the performance of individual modules.

## 6. Conclusions

This paper presents and describes the architecture of a novel application for AR navigation in inland and coastal waters. While the proposed architecture partially relies on existing, well-known technological solutions, the overall concept constitutes remarkable advancements with respect to existing systems. This research aimed to show the current state of the art of existing AR solutions for nautical applications and propose a methodology to introduce AR for mainstream mobile devices for shallow water navigation and nautical tourism development. The proposed solution is the first step to test AR implementation's feasibility and propose future AR evolution in this field.

The presented solution also proposes a participative approach where sailors can become part of the data acquisition process. They are a valuable resource providing information on hazard locations, traffic information, current weather conditions and much more. This allows increasing up-to-date AR contents dynamically. Due to validation issues

with data provided using crowdsourcing, it needs to be verified using strict validation procedures before displaying.

The concept of the proposed AR solution has already been set up as a working prototype of the AR inland water app of the Masurian Lakeland region in Poland with several operative modules. The deployment of the whole vision will require long-term work due to the need to integrate a wide range of different technical solutions. The authors will continue this work to support safety in the navigation process on inland and coastal water with AR technology.

Currently, there is no representative standard of navigation information elements related to AR technology. This issue has yet to be investigated and tested on end-users to define representative guidelines. As research and development progresses, various navigation aid systems based on different displays will be further developed and utilised to increase the immersion level of the provided AR content. It is expected that this application of augmented technology will prevent future accidents on the water and provide new content.

In conclusion, the presentation of Augmented Reality in an aquatic environment is a complicated and challenging task. Compared to other industry sectors, this AR application is still underdeveloped. This shows that additional research is needed to leverage this technology's potential fully. Additionally, tools and processes need to be optimised for efficiency.

The study showed that further research is needed on the technical side of the platform. There are options to extend the platform with external sensors, cameras, and miniaturised scanners or radars to provide real-time data that can be analysed using artificial intelligence, object detection, collision avoidance, and even autonomous movement.

**Author Contributions:** Conceptualization, T.T., D.P., and M.G.; methodology, T.T., D.P., and M.G.; validation, T.T. and M.G.; formal analysis, T.T. and M.G.; investigation, D.P.; resources, T.T. and D.P.; writing—original draft preparation, T.T.; writing—review and editing, T.T. and D.P.; visualisation, T.T. and M.G.; funding acquisition, D.P. All authors have read and agreed to the published version of the manuscript.

**Funding:** This research was founded by: University of Warmia and Mazury, Olsztyn, Poland, and ESA Contract No. 4000127827/19/NL/MM/ff.

**Institutional Review Board Statement:** Not applicable.

**Informed Consent Statement:** Not applicable.

**Data Availability Statement:** Not applicable.

**Acknowledgments:** The authors wish to thank INFEO company for support and giving us access to the system data.

**Conflicts of Interest:** The authors declare no conflict of interest.

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
