# Peer review of "Using Augmented and Virtual Reality (AR/VR) to Support Safe Navigation on Inland and Coastal Water Zones"

_remotesensing, doi:10.3390/rs14061520_

Round 1

Reviewer 1 Report

Electronic navigational systems are used in marine, aerial and land navigation. They integrated navigational sensors for presentation them in one place and visualisation the navigational situation of own vehicle. First they were used in marine navigation as ECSs (electronic chart systems) and ECDISs (electronic chart display and information systems) presenting on raster or vector chart own position and moving vector (ECS) enriched with information about navigational marks, obstacles et al (ECDIS). Other parameters can be presented (on the basis of own sensors): wind power and direction, the depth, radar and ARPA objects, AIS information.

I the article "Using Augmented and Virtual Reality (AR/VR) to support safe navigation on inland water" the Authors propose a new solution to assist sailors in safe navigation on inland shallow waters by using Augmented and Virtual Reality. In inland areas with GSM operational zone, there is possible to obtain an information about object loggged in the service to obtain weather forecast and warnings and exchange an information about other vessels/yachts. In car systems an information about trafic intensity is enabled. The structure of the article is considered and clear. The background and comprehensive review of the problem's literature were presented. Discussion and conclusions, on the basis of the research, are comprehensive and clear.

Following suggestions should be taken into consideration:
Line 175: A ship on water is not static but can roll, pitch, and yaw. Please note, it moves and there is a heave. Yaw is a result of changing the course and moving disruptions are: roll, pitch and heave measured in MRU (motion referece unit).
Line 227: Navigation maps used in marine navigation are called "charts".
Line 258-271: These parameters are presented in ECS and ECDIS for 25 years or more (but AIS - about 15-20 years)
Line 319: EGNOS is a SBAS not satellite system and does not provide subcentimeter accuracy.
Line 381: Ar-optimised should be corrected to be in capital letters (AR-optimised)
Line 411 and 416: an information, that Bełdany lake is very popular is written twice
Line 423: horizontal location should be rather used than spatial

Figure/figure should be standarized - rather in capital letter as in the instruction for Authors
Abbrefiations should be described when first mentioned: IALA, POI, MOPR.

The article is very interesting and has high scientific advantages.
I recommend this manuscript for publication in Remote Sensing after text editig.

Author Response

Dear reviewer

We would like to thank for sparing the time to write detailed and useful comments. Following the comments, we have made all the suggested changes. The all changes in the manuscript have been tracked.

Kind regards,

Authors

Reviewer 2 Report

The article in question addresses the basic challenge in marine navigation: it is not easy always easy to identify what is what within the observed area. The capability to note relevant information on sea can be called ‘seaman’s eye’ (Crenshaw, R. 1965, Naval Shiphandling, US Naval Institute), a skill that develops through experience. An AR system could then, indeed, provide the user with artificially enhanced ‘seaman’s eye’ by clearly indicating the to-be-identified targets (such the port to which the ship is to be sailed) and also important elements that are invisible to the naked eye, underwater rocks and shallow areas in particular. From this perspective, the paper “Using Augmented and Virtual Reality (AR/VR) to support safe navigation on inland water” is worth publication. I also find the article well written (but please note that I’m not a native English speaker). Although the user study part is documented rather briefly, the article seems the feature good, although not perfect, understanding of marine navigation. There are certain considerations, which should be noted for enhancing the quality of the paper.

Firstly, the article addressed the issue of navigation on inland waters. More specifically, the test area is the Great Masurian Lakes, which is described as challenging navigational area. It is somewhat unclear what is meant with “up-to-date bathymetric surveys” [line 402], whether this is bi-annual or annual or even more common. Up-to-date maps are important for navigation on any challenging marine area, but perhaps even more important they are at the Great Masurian Lakes? In any case, the problem with the rationale here is that there are certain sea areas that can be rockier and more congested than certain inland waters, although they might not be as common. For example, archipelagos around Stockholm, Helsinki and Turku are more challenging, I would assume, than certain large lakes. From this perspective you could consider re-naming your article, since an efficient AR-solution could be equally useful at coastal areas as well.

Indeed, you write that “[a]ccording to the information available to the authors, there is no solution dedicated to ordinary users navigating inland water reservoirs. Existing solutions focus on implementing AR within professional, dedicated marine vessel solutions.” [lines 661-662]. This contrast is problematic, because some professionals navigate through inland waters. Perhaps you could frame your paper in a manner such that you are developing an AR solution for non-professionals (who still have not developed their ‘seaman’s eye’) navigating on challenging sea areas. A key difference between professionals and leisure sailors is that, usually (though not always), the professionals tend to observe the marine environments behind windows, which might allow HUD-solutions. You, in turn, feature a smart phone app solution, an approach that could be well suited for non-professional sailors with less budget and fewer possibilities for HUD-system. Perhaps there is also the possibility that in not too distant future, suitable AR glasses will be developed for non-professional budget.

Furthermore, it might be that a pad or smartphone solution could be useful for non-professionals in particular. A leisure seafarer might be interested on such solution for various reasons. Here, however, the AR solution should compete with and complement the regular digital map view solutions. The AR solution might have some benefice here. Firstly, and perhaps least importantly, digital maps apply GNNS, which, in principle, is not always reliable. The AR solution might apply camera and machine vision for geolocation purposes: I do not note that machine vision based celestial navigation would have been mentioned in your article (which, however, might not work on cloudy days).

Secondly, a smartphone app could simply serve in providing added certainty for novice seafarers. Even with digital map and GNNS-based location, it is sometimes uncertain what is what due to underdeveloped ‘seaman’s eye’. In such case, the digital map and the smartphone app could be used in conjunction: one could click on the map, say, a rock or small island, and then that object would be highlighted on the AR application. To the best of my understanding, this simple solution was not discussed in your article draft. You could also further discuss the importance of bathometry for non-professionals: it is indeed so that shallow areas should be avoided, but 2D digital map can be extremely efficient for such purpose. Perhaps AR-based bathometric presentations could be useful for diving or fishing purposes.

Thirdly, unlike professionals, the leisure seafarers might have underdeveloped sense of vessel trajectories. Although professionals are able to read the overall marine traffic and have eye for future locations of the other vessels, the other ships are nevertheless the main source of uncertainty. Perhaps in the future waypoints will be automatically shared between ships for increased certainty in avoiding collisions. Other vessels are a crucial challenge for leisure seafarers as well. It is notable that within the present regulation, small boats needn’t to have send AIS (automatic identification system) data and are therefore not visible on the digital charts. Perhaps AR solution with image recognition capabilities might help identify other vessels and predict their future locations based on tracking movements and calculating trajectories: predicted future locations could be presented on the AR field of vision. However, such solution could be technically challenging to achieve. Alternatively, automatic exchange of information could be easily achieved, if both leisure vessels are users of the same app, akin to as you have proposed. However, and this is a crucial weakness, I did not note that you would have proposed that the vessels share information relevant for navigation. These include future locations and more generic vessel intentions and conditions that would help in anticipate the marine traffic. Sharing waypoints might not be possible, if such navigational system is not in use or integrated on the proposed app. Nevertheless, such app one could at least be used for signalling issues that are usually presented via maritime signal flags (such as, diver down) or other markers, such as “day shapes” or lights (such as when using a ball signal for telling the others that you are anchored). Such information could be presented on the AR vision. Overall, lack of consideration to “anticipatory marine navigation” (Wahlström, M., Forster, D., Karvonen, A., Puustinen, R., & Saariluoma, P., 2019, Perspective-taking in anticipatory Maritime navigation – Implications for developing autonomous ships) is a weakness in your consideration to the potential usefulness of AR for future seafarers.

Hopefully, these considerations help in slightly reframing the paper and enhancing the discussion section at least. I especially recommend 1) to reconsider framing of the paper, i.e., in my opinion this paper is not about inland water solutions, because there is not such big difference between inland and coastal navigation (both include professionals and non-professionals as well as challenging and easier environments). Secondly, 2) I would expand the description on marine navigation and the challenges involved; the sources mentioned above, or similar alternatives, should be used in this. Thirdly, 3) I would expand on the possibilities of AR in addressing the said challenges within marine navigation.

Author Response

Dear reviewer

At the beginning, I would like to thank you for your insightful and extremely valuable review. The discussion on the usefulness of AR tools in supporting the process of safe navigation among experienced sailors as well as occasional, tourist sailors is very factual and interesting.

This article is about navigation on inland waters, but the problem and solution can be applied to rivers and lakes as well as to coastal waters. In our research we have focused on the closest body of water to us where we can conduct research. However, the problem of the validity of the bathymetric map, the problem of navigational obstructions occurs in both the inland and coastal test areas. I fully agree that our solution, and therefore the title and content of this article, can be extended to include coastal waters. So, we did it.

Lines [661-662]. We propose to remove the sentence “According to the information available to the authors, there is no solution dedicated to ordinary users navigating inland water reservoirs.” We have further made minor amendments to this paragraph. We added sentence “At the same time there is a large group of sailors with less experience navigating on challenging sea and shallow areas. There are also professionals who sail recreationally in shallow waters.”

The suggestion that AR solution using camera and machine vision for geolocation could be used is very interesting and to be considered for further research.

We, agree, that 2D digital map can be extremely efficient for navigation purpose but AR-based bathometric presentations could be useful for diving or fishing purposes.

We are thinking of introducing a system for users/vessels to alert each other. The easiest way would be to introduce this as a first step within the same app. Users already see other boats logged into the same application. Nevertheless, this is another big step in the development of the application, it requires additional work and tests, and it is difficult to put it in one article.

Thank you also for your suggestions to consider the publication “anticipatory marine navigation”. We will take it into account during further development of the application.

Following the comments, we have made the suggested changes. The all corrections in the manuscript have been tracked.

Thank you once again for your valuable suggestions.

Kind regards,

Authors

Reviewer 3 Report

Interesting topic for marine operations and remote sensing.

Virtual Reality and VA are becoming more low cost are easier to buy and use by common users. Therefore, it will increase the use of these tools in several areas.

The use of VR for support safe navigation on inland water is a good engineering development.

I'm a sailor and I know the difficulties to track COLREG rules. Use of VR/VA systems on nautical water ways is critical and a great topic.

Author Response

Dear reviewer

We would like to thank for sparing the time to analyse the manuscript. Following the comments, we have made all the suggested changes. The all changes in the manuscript have been tracked.

Kind regards,

Authors